# Evaluating the Functional Properties of Spray-Dried Porcine Plasma in Gilthead Seabream (*Sparus aurata*) Fed Low Fish Meal Diets

**DOI:** 10.3390/ani12233297

**Published:** 2022-11-25

**Authors:** Laura Fernández-Alacid, Ignasi Sanahuja, Cristina Madrid, Javier Polo, Joana P. Firmino, Carlos Balsalobre, Felipe E. Reyes-López, Eva Vallejos-Vidal, Karl B. Andree, Enric Gisbert, Antoni Ibarz

**Affiliations:** 1Department of Cell Biology, Physiology and Immunology, Faculty of Biology, University of Barcelona, Avda. Diagonal 643, 08028 Barcelona, Spain; 2IRTA, Centre de La Ràpita, Aquaculture Program, 43540 La Ràpita, Spain; 3Department of Genetics, Microbiology and Statistics, Faculty of Biology, University of Barcelona, Avda. Diagonal 643, 08028 Barcelona, Spain; 4APC Europe SL, Avda. Sant Julià 246-258, 08403 Granollers, Spain; 5Fish Health and Integrative Physiogenomics Research Team, Center of Biotechnology and Aquaculture, Faculty of Chemistry and Biology, University of Santiago de Chile, 9170002 Santiago, Chile; 6Núcleo de Investigación Aplicada en Ciencias Veterinarias y Agronómicas, Facultad de Medicina Veterinaria y Agronomía, Universidad de Las Américas, 8242125 Santiago, Chile

**Keywords:** aquaculture, functional feed, immunity, skin mucus-associated biomarkers (SMABs), spray-dried porcine plasma (SDPP)

## Abstract

**Simple Summary:**

Spray-dried porcine plasma (SDPP) is considered one of the most promising functional ingredients as an alternative to the use of antibiotics, due to growth and immunity promotion. In the current work, we analyzed a mucosal immune response through challenging skin mucus antibacterial capacity and a systemic immune response through their splenocytes ex vivo with a pathogen-associated molecular pattern. Main skin mucus biomarkers were not modified by the SDPP diet, except for cortisol levels that were drastically reduced, which could be considered beneficial from a physiological perspective. In addition, the SDPP diet significantly reduced the growth of the pathogenic bacteria studied in the skin mucus. Anti- (*tgf-β1*), and pro-inflammatory (*il-1β*) cytokines, genes involved in humoral immunity (*IgM*), and the surface cell marker cd4 (*cd4*) were up-regulated in primary cell cultures of splenocytes exposed to lipopolysaccharide in fish fed the SDPP diet, which indicated that this functional ingredient was able to modulate the systemic immune response. In addition to promoting somatic growth, this study also indicated that SDPP, especially in the current low fish meal diets, seems to be useful and beneficial at the immunological level and mucosal health status, which makes it a good candidate for incorporation into the aquaculture sector.

**Abstract:**

Blood by-products are an untapped source of high-quality ingredients for aquafeeds, containing a broad variety of cytokines, hormones, growth factors, proteins, bioactive peptides, and amino acids. The effects of the spray-dried porcine plasma (SDPP), a type of processed animal protein on several immune parameters, were evaluated in sea bream using ex vivo and in vitro assays. In this study, fish were fed with two isoproteic, isolipidic, and isoenergetic diets: control diet (7% fish meal, FM) and SDPP diet (2% FM and 5% SDPP). At the end of the 92-days trial, those fed the SDPP diet were larger in body weight (*p* < 0.05) without differences in feed conversion ratio (*p* > 0.05). The ex vivo immune stimulation of splenocytes indicated that SDPP had a beneficial effect in promoting systemic immunity, since the surface cell marker (*cd4*), pro- (*il-1β*), and anti-inflammatory (*tgf-β1*) cytokines, and genes involved in humoral immunity (*IgM*) were up-regulated. The co-culture assays of skin mucus corroborated that SDPP enhanced the antibacterial capacity of mucus against *V. anguillarum*. In addition, main mucus biomarkers did not show significant differences, except for cortisol levels which were lower in the SDPP diet. The present study indicated that SDPP may be considered a functional ingredient in aquafeeds formulated with low FM levels.

## 1. Introduction

The formulation of feeds promoting fish health, immunity, and welfare without compromising somatic growth or product final quality, has become a reality in the last years [1]. Nowadays, aquafeed companies are betting heavily on the development of so-called functional feeds. This change of paradigm has been mainly driven by societal criticism of the prophylactic use of antibiotics and chemotherapies that have eventually led to their ban in many countries [2]. Thus, the inclusion of dietary functional ingredients and feed additives that can promote the immunological condition of cultured aquatic species has received a lot of attention from feed manufacturers [2,3,4]. This is of special relevance in diets containing high levels of plant protein sources. For instance, even though different studies have reported that diets with high levels of fishmeal replacement by plant protein sources may be formulated without negatively impacting on growth and feed performance parameters, they have been reported to depress fish immunity [5,6,7].

A by-product of the slaughter industry, the spray-dried plasma (SDP), is obtained from healthy poultry or porcine whole blood after anticoagulant treatment, erythrocyte extraction by centrifugation, and dehydration through the spray-drying technique [8]. In livestock production, SDP is considered a promising functional ingredient as an alternative to the use of antibiotics, due to growth and immunity promotion, as well as reducing both morbidity and mortality [9,10,11]. Regarding aquatic species, several studies have reported that spray-dried porcine plasma (SDPP) enhances growth in rainbow trout (*Oncorhynchus mykiss*) [12], gilthead sea bream (*Sparus aurata*) [13,14,15], and Nile tilapia (*Oreochromis niloticus*) [16]. Different authors have postulated that these results might be explained by the higher digestibility of SDPP, the improvement of feed efficiency and feed intake, as well as its content in growth factors. In addition to its use as a source of highly digestible dietary protein source, SDPP has also been recommended in livestock and aquaculture feeds as a source of immunological support due to its immune modulatory components, including immunoglobulins and bioactive peptides [10,17]. In particular, aquafeeds containing SDPP enhanced the local gut and systemic immunity in sea bream [13,15,18], leukocyte, lymphocyte, and neutrophil counts in tilapia [16], and skin mucus antibacterial capacity in meagre *Argyrosomus regius* [19]. Although the benefits of SDPP in livestock diets are well known, there is limited literature on the immunomodulatory effects of its dietary inclusion in aquatic feeds, especially when fish may be challenged by different biotic and abiotic stressors.

Thus, using as a biological model the gilthead seabream, the objective of this study was to evaluate the beneficial effects of SDPP in diets formulated with very low fishmeal (2%) in terms of standard key performance indicators like somatic growth and feed efficiency parameters and test whether diets containing SDPP promoted the systemic immune response in gilthead sea bream by: (a) a mucosal immune response through challenging skin mucus antibacterial capacity in order to evaluate the response of the skin-associated lymphoid tissue in front of bacterial pathogens; (b) a systemic immune response through their splenocytes ex vivo with a pathogen-associated molecular pattern like lipopolysaccharide (LPS) and evaluate the effects of the functional ingredient on the systemic immune response of the host.

## 2. Materials and Methods

### 2.1. Animals, Diets, and Sampling

Gilthead sea bream juveniles were purchased from a fish farm (PISCIMAR SL, Andromeda Group, Castellón, Spain), transported by road to IRTA facilities in La Ràpita. After their acclimation, fish (body weight, BW = 26.0 ± 4.0 g, mean ± standard deviation, SD) were distributed in eight 450 L tanks (35 fish per tank) connected to a water recirculation unit IRTAmar^®^(La Ràpita, Spain) working under open-flow conditions. Fish were fed two diets for 92 days (4 replicates per diet) under the following environmental conditions: water temperature values ranging from 22 to 27 °C, 6.1 ± 0.2 mg/L of dissolved oxygen (OXI330, Crison Instruments, Barcelona, Spain), and 7.5 ± 0.01 pH (pH meter 507, Crison Instruments), and natural photoperiod (June–September; latitude 40°37′41″ N). Feeds were distributed four times per day by automatic feeders (ARVO-TEC T Drum 2000^TM^, Arvotec, Huutokoski, Finland), at the rate of 3.3% of the stocked biomass, which approached apparent satiation.

A control diet was formulated (0% SDPP) with low levels of marine-derived protein sources (7% fish meal 70LT, Norvik, UK) with high inclusion of a blend of vegetal protein sources that fulfill the known nutritional requirements of juvenile sea bream. Based on this basal formulation, an additional diet was formulated, in which SDPP (APPETEIN™ GS; APC Europe, SL, Granollers, Spain) was incorporated at 5% [13], at the expense of high-quality fishmeal. Both diets were isoproteic (48% crude protein), isolipidic (17% crude fat), and isoenergetic (21.6 MJ/kg gross energy) (Table 1). Diets were manufactured by temperature-controlled extrusion by Sparos Lda. (Olhão, Portugal) [13].

This study was conducted following the Guiding Principles for Biomedical Research Involving Animals (EU2010/63), the guidelines of the Spanish laws (law 32/2007 and RD 53/2013) and authorized by the Ethical Committee of the Institute for Research and Technology in Food and Agriculture (Spain) for the use of laboratory animals and Generalitat de Catalunya (FUE-2020-01809522).

At the end of the trial, sampling took place to monitor fish growth. For that purpose, all fish in each tank (4 replicates per diet) were netted and anesthetized with tricaine methane sulfonate (MS-222, 150 mg/L), and their BW (g) and standard length (SL, cm) were measured to the nearest 0.1 g and 1 mm, respectively.

The following key performance indicators associated with somatic growth and feed efficiency were measured at the end of the trial: BW, SL, Fulton’s condition factor (K), daily weight gain (DWG), specific growth rate (SGR) and feed conversion ratio (FCR). The specific growth rate (SGR) was measured using the formula: SGR (% BW day^−1^) = (ln BW_f_ − ln BW_i_) × 100/days, and the Fulton’s condition factor (K) was measured using the formula: K = (BW_f_/SL_f_^3^) × 100. BW_f_ and BW_i_ were the final and initial BW in grams, and SLf the final SL in cm. Feed utilization was evaluated by the following formula: feed conversion ratio (FCR) = feed intake (g)/increase in fish biomass (g). Fish survival was also recorded.

### 2.2. Skin Mucus Procedures

#### 2.2.1. Skin Mucus Collection

At final sampling, 40 animals were randomly selected (20 fish for each dietary condition; 5 fish per tank) for mucus analyses. For this purpose, animals were anaesthetized with MS-222 (100 mg L^−1^) and skin mucus was collected as described in Fernández-Alacid et al. [20].

#### 2.2.2. Skin Mucus Associated Biomarker Analyses

The Bradford assay [21] was used to determine the soluble protein concentration of homogenized skin mucus samples, using the bovine serum albumin as a standard (BSA; Sigma-Aldrich, Madrid, Spain). Protein values were expressed as mg mL^−1^ of skin mucus. Glucose and lactate concentrations were determined by commercial enzymatic colorimetric tests (SPINREACT^®^, Barcelona, Spain) following the manufacturer’s instructions but with slight modifications [20]. Glucose and lactate values were expressed as μg mL^−1^ of mucus and μg mg^−1^ of skin mucus protein. Skin mucus cortisol levels were measured using an ELISA kit (IBL International, Hamburg, Germany) as previously described in Fernández-Alacid et al. [20]. Ferric reducing antioxidant power (FRAP) detection was determined by an enzymatic colorimetric test (Invitrogen, Thermo Fisher Scientific, Barcelona, Spain). Briefly, 20 μL of mucus or standard solutions (FeCl_2_) was mixed with 75 μL of FRAP color solution and incubated for 30 min at room temperature. Antioxidant values were expressed as nmol FRAP mL^−1^ of mucus, and nmol mg^−1^ of skin mucus protein.

#### 2.2.3. Skin Mucus Antibacterial Capacity Evaluation by Co-Culture Challenges

Three different bacteria for marine fish were tested to study the skin mucus antibacterial capacity: *Escherichia coli* (DSMZ423) a non-pathogenic bacteria, and two pathogenic bacteria like *Vibrio anguillarum* (CECT522T), and *Pseudomonas anguilliseptica* (CECT899T). The different strains were grown in Tryptic Soy Broth culture media (TSB, Conda, Spain), and in Marine Broth culture media (MB, Difco Laboratories, Detroit MI, USA), respectively. The impact of skin mucus on strain viability was evaluated by monitoring the absorbance of co-culture grown in 96-well plates [22].

### 2.3. Ex Vivo Procedures

#### 2.3.1. Ex Vivo Immune Stimulation of Splenocytes with LPS

Six randomly selected animals from each experimental group (biological replicates) were sacrificed with an overdose of MS-222 as previously described, and their spleens were removed. The detailed description of the ex vivo protocol in which splenocytes were challenged with LPS may be found in Campoverde et al. [23].

#### 2.3.2. RNA Extraction, cDNA Synthesis, and Gene Expression Analyses

At 0, 4, 12, and 24 h of the splenocytes incubation with PBS (control) or LPS, total RNA was extracted from cell cultures using the QIAGEN RNeasy^®^ Mini Kit (Barcelona, Spain) following the manufacturer’s recommendations. RNA concentration was determined by spectrophotometry with a ND-2000 NanoDrop (Thermo Scientific™, Barcelona, Spain) and its quality evaluated by means of agarose (2%) gel electrophoresis. Then, single-stranded DNA (cDNA) was synthesized to quantify the expression of the genes under study. For cDNA synthesis, 1 μg of total RNA was transcribed using a high-capacity cDNA reverse transcription kit (QuantiTect^®^ Reverse Transcription Kit, Barcelona, Spain) following the instructions provided by the manufacturer.

Gene expression analyses by qRT-PCR from splenocytes’ primary cell cultures were similar to those already described by Salomón et al. [24]. The following genes were analyzed considering their immune function: lysozyme (*lys*), immunoglobulin M (*igM*), interleukin 1 beta (*il-1β*), tumor necrosis factor alpha *(tnfα*), interleukin 10 (*il-10*), *transforming growth factor beta1 (tgfβ1*), the surface cell marker cd4 (*cd4*), manganese superoxide dismutase (*mn-sod*) and catalase (*cat*). β-Actin was included in our analysis as the reference gene based on BestKeeper© software [25]. The specific primers set for each gene are detailed in Salomón et al. [26], whereas primer efficiency was calculated according to Pfaffl [26]. qRT-PCR reaction conditions may be also found in Salomón et al. [24]. Gene expression values were expressed as normalized relative expression (NRE) and were calculated in relation to values of the control group at 0 h and normalized against those of the reference gene. The results were expressed as mean expression values obtained at 0, 4, 12, and 24 h of incubation with PBS or LPS (*n* = 6 fish per diet, experimental condition, and time-point assessed).

### 2.4. Statistical Analysis

Results are expressed as mean ± standard deviation (SD), except for data related to the skin mucus antibacterial activity against different bacteria were plotted as mean ± standard error (SE). All data were checked for normality and homoscedasticity prior to their analysis. Differences in somatic growth and skin mucus parameters between both groups were compared by means of a student’s *t*-test, whereas the two-way ANOVA test was used to determine differences in gene expression between both diets and sampling times. When no statistical interaction was detected between the two tested factors (time and diet) by the two-way ANOVA, results were treated as a one-way ANOVA. In both cases, when statistically significant differences were detected among groups (*p* < 0.05), the ANOVA was followed by the post-hoc Bonferroni test. Statistics were done using Graph Pad Prism V.6.1. (GraphPad Software, San Diego, CA, USA).

## 3. Results

### 3.1. Somatic Growth

At the end of the 92-days trial, fish fed the diet containing 5% SSPP at the expense of FM were 3.8% heavier (180.9 ± 4.4 g) than those fed the control diet 173.8 ± 4.1 g (Table 2; *p* < 0.05). No significant differences in final SL (18.7 ± 0.2 cm vs. 19.0 ± 0.3 cm, respectively) and K (2.65 ± 0.1 vs. 2.64 ± 0.2. respectively) were found between both groups (*p* > 0.05).

### 3.2. Skin Mucus Biomarkers Level and Antibacterial Activity

The main skin mucus-associated biomarkers (SMABs) related to metabolism, stress and antioxidant power are presented in Table 3 for both tested diets. No significant statistical differences were found among dietary groups in terms of the exuded mucus protein, glucose, and lactate contents (*p* > 0.05). However, mucus cortisol levels were significantly decreased by 40% in fish fed the SDPP diet in comparison to the control group (*p* < 0.05), whereas no modifications were observed for the mucus antioxidant power (FRAP) (*p* > 0.05). Similar results were found when data were expressed as relevant ratios (Table 3), except for FRAP levels that were reduced to 20% in the SPDD diet.

The impact of the SDPP diet on the skin mucus antibacterial activity against *E. coli*, *V. anguillarum*, and *P. anguilliseptica* in a co-culture challenge is shown in Figure 1, which depicts their respective bacterial growth (Figure 1A,C,E) and the inhibitory capacity of the skin mucus (Figure 1B,D,F). Regarding the non-pathogenic bacteria *E. coli*, the skin mucus co-culture (14 h of incubation) did not show differences between the control and the SDPP diets (*p* > 0.05), with similar maximum inhibitory power between the 4 and 8 h of co-culture period (Figure 1B). The mucus antibacterial capacity against the pathogenic bacteria *V. anguillarum* was enhanced in the SDPP diet through all of the co-culture period with respect to control diet (Figure 1B; *p* < 0.05), resulting in a significantly higher inhibitory power (Figure 1D). Skin mucus from both dietary groups evidenced a great capacity to inhibit *P. anguilliseptica* growth, reaching inhibitory values of 76% after 10 h of co-culture. Although growth curves for both diets were similar, the SDPP diet significantly improved inhibitory power at 14 h of the co-culture period (Figure 1F).

### 3.3. Gene Expression Analyses in Splenocytes from the Ex Vivo Assay

The gene expression of humoral innate response markers measured in primary cultures of splenocytes incubated with PBS and LPS from both experimental groups are shown in Figure 2. In particular, no interaction between tested factors (diet and time) were detected by the two-way ANOVA (*p* > 0.05). The expression values of *lys* in splenocytes from fish fed both diets and incubated with PBS remained stable at a basal level over the 24 h experimental period (*p* > 0.05). The incubation with LPS stimulated *lys* expression at 12 h in splenocytes from fish fed both experimental diets when compared to PBS (*p* < 0.05). At 24 h post LPS stimulation, the expression of *lys* for the control diet was similar between cells exposed to PBS and LPS (*p* > 0.05). In contrast, fish fed with the SDPP diet showed higher *lys* expression values in splenocytes exposed to LPS at 12 h and 24 h (*p* < 0.05), indicating a sustained expression of this humoral gene marker over time.

Expression of *IgM* values in splenocytes from the control group incubated with LPS remained constant over the 24 h exposition period (*p* > 0.05). In contrast, *IgM* expression levels differentially increased at 4 h in splenocytes from fish fed with the SDPP diet when exposed to LPS compared to PBS (*p* < 0.05). Such increase in the *IgM* expression was maintained at 12 and 24 h (*p* < 0.05). Thus, the SDPP diet improved the *IgM* expression compared to the control diet when the cultured splenocytes were exposed to LPS (*p* < 0.05).

The *tnfα* expression value showed an earlier upregulation in splenocytes from both dietary groups incubated with LPS. Thus, the *tnfα* expression levels increased at 4 h when exposed to LPS (*p* < 0.05) to then decreased at 12 h, achieving basal expression levels similar to those recorded at 0 h (*p* > 0.05). On the other hand, the *tnfα* expression in splenocytes from fish fed with both diets and incubated with PBS remained stable over the 24 h experimental period (*p* > 0.05). 

The expression of *il-1β* registered a marked upregulation at 4, 12, and 24 h, especially in the splenocytes from fish fed with SDPP and stimulated with LPS. In line with the other gene expression values, the *il-1β* remained constant along the 24 h experimental period (*p* > 0.05) in splenocytes from fish fed both experimental diets and incubated with PBS. The expression levels of *il-1β* in splenocytes from both experimental groups increased at 4 h when incubated with LPS in comparison to PBS (*p* < 0.05). This increase was sustained at 12 h (although in a lower magnitude but still high) and showed no differences between the control diet- and SDPP diet-splenocytes (*p* > 0.05). Importantly, at 24 h no differences were registered in *il-1β* between splenocytes from the control group when incubated with PBS or LPS (*p* > 0.05), but the upregulation of *il-1β* for the SDPP group persisted (*p* < 0.05). In addition, *il-1β* levels at 24 h were significantly higher in splenocytes from fish fed with the SDPP diet compared to those from the control group (*p* > 0.05).

The expression values of the surface marker *cd4,* anti-inflammatory cytokines (*tgfβ* and *il-10*), and antioxidative stress enzymes catalase and superoxide dismutase were also evaluated (Figure 3). No interaction between tested factors (diet and time) were detected by the two-way ANOVA (*p* > 0.05). For *tgfβ*, no statistically significant differences were found between groups nor sampling times regardless of the incubation of splenocytes with PBS or LPS. Only at 24 h the splenocytes from the SDPP group exposed to LPS showed higher expression values compared to PBS-incubated splenocytes (*p* < 0.05) and compared to the control diet exposed to LPS (*p* < 0.05). Transcription levels of *il-10* showed a transient increase at 12 h was observed in the LPS-incubated splenocytes from control and SDPP diet (*p* < 0.05) compared to the PBS-incubated splenocytes (control and SDPP diet). At 24 h post-incubation, expression levels decreased and were similar to those recorded at 0 h (*p* > 0.05). Regarding levels of *cd4* in both experimental groups (PBS- and LPS-incubated splenocytes) expression remained constant up to 24 h; however, at this sampling time, *cd4* expression values in SDPP splenocytes incubated with LPS were higher than those in their counterparts incubated with PBS (*p* < 0.05). Similarly, *cd4* expression levels in splenocytes incubated with LPS from the SDPP group were higher than those measured in the control group (*p* < 0.05).

Regarding expression levels of antioxidative stress enzymes, *sod* expression values increased between 0 and 4 hours post-incubation in all conditions and remained stable until the end of the study. At 24 h, *sod* expression levels decreased in splenocytes from the control group incubated with LPS in comparison with those with PBS (*p* < 0.05). By contrast, there were no differences between *sod* expression values in SDPP splenocytes incubated with PBS or LPS (*p* > 0.05).

Similar to *sod*, transcription levels of *cat* slightly increased from 0 to 4 h and remained stable until the end of the study. No major changes in gene expression values were found at 4, 12 and 24 h except for samples incubated with LPS from the control group that showed lower expression values in comparison to those from the same group incubated with PBS and those to the SDPP group incubated with LPS (*p* < 0.05).

## 4. Discussion

The term “functional feed” has arisen as a nutritional strategy to improve the general health status, including welfare and stress in farmed aquatic organisms [1,27]. Among different animal by-products, SDPP included in animal diets benefited systemic health and mucosal performance both in livestock [10] and aquaculture species [15,18]. Blood by-products are an untapped source of high-quality ingredients for aquafeeds, containing a broad variety of amino acids, proteins, hormones, growth factors, bioactive peptides, and cytokines [10,28]. In this study, the inclusion of SDPP at 5% in diets with low FM levels (2%) had a beneficial effect on somatic growth as values on BW and SGR indicated. These results are similar to those already reported in other studies dealing with this ingredient and could be attributed to the high apparent digestibility values of porcine blood by-products derived from plasma [29,30,31], as well as their content in biologically active peptides and growth factors [17,32].

As it has been previously stated, functional feeds are commonly used to modulate the immune and stress responses; however, predicting functional response is very difficult due to complex interactions between dietary ingredients [33]. Although there are benefits of SDPP in terms of growth, feed efficiency and host’s immunity in diets formulated with high levels of FM (46%) [13,15,18], there is not information on whether low FM diets (2%), such as those tested in the current study and in our previous studies [14], would behave similarly to full FM ones. Thus, in this study, the authors aimed to evaluate the potential immunomodulatory effect of SDPP inclusion in diets with very low levels of FM (2%) by means of an ex vivo assay. For this purpose, the splenocytes from both experimental groups were stimulated with bacterial LPS and an in vitro challenge to evaluate the skin mucus, antibacterial capacity was also assessed using a co-culture system. The ex vivo assay revealed that SDPP enhanced the immune response of juvenile gilthead sea bream, as indicated by the up-regulation of the gene markers involved in the humoral innate response like the *igM*, as well as pro-inflammatory cytokines such as *il-1β*, anti-inflammatory cytokines like the *tgfβ1*, and the surface cell marker cd4 (*cd4*) at 24 h post-exposure to LPS. These findings in gilthead sea bream are supported by data on piglets where SDPP decreased both pro-inflammatory and anti-inflammatory cytokines in the intestinal mucosa, which the authors attributed to a reduction in blood monocytes, macrophages, and B and T lymphocytes in gut lymphoid tissue [34,35,36].

Immunoglobulins are key players in orchestrating the innate and adaptative immunity. In fish, the most abundant immunoglobulin in plasma and skin mucus is *IgM* [37] that plays an important function in systemic humoral immune responses [38]. The increased *igM* expression levels in splenocytes from animals fed the SDPP diet at 4-, 12- and 24-h post-incubation (hpi) in LPS may indicate that the SDPP diet improves the *IgM* expression compared to the control diet, as it has been previously shown by several studies testing functional feeds [14,39,40,41]. Moreover, the ex vivo assays showed an up-regulation of *il-1β* at 24 hpi in splenocytes exposed to LPS from fish fed the SDPP diet. This is particularly relevant since the *il-1β* cytokine is responsible for a cascade of effects on different members of this pro-inflammatory family in fish, having a key role in signal transduction and activation of the nuclear factor (NF)-κB pathway; thus, participating in the regulation of inflammatory responses, apoptosis, and cellular growth [42]. Thereby, IL-1β in fish species is analogous to mammals and it is involved in the regulation of immunity through the stimulation of T cells [43]. Accordingly, the upregulation of *cd4* expression at 24 hpi in splenocytes incubated with LPS from fish fed the SDPP diet may be related to the activation and differentiation of T helper cell subsets [44]. Other cytokines involved in adaptative immunity and present in fish, may form the effectors of potential T regulatory cells, as such as IL-10 and TGF-β1. Under the above-mentioned ex vivo conditions in splenocytes exposed to LPS at 24 hpi, the expression levels of the anti-inflammatory cytokine *tgfβ1* were higher from fish fed the SDPP diet in comparison to the control group. Other nutritional studies when evaluating the functional properties of plasma proteins in mice [35] and piglets [45] have found the upregulation of anti-inflammatory cytokines like in the current study. In this sense, the expression of *tgfβ1* corroborated the anti-inflammatory properties of the plasma proteins included in SDPP ahead of the initial acute inflammatory response. This may support the hypotheses that SDPP reduces over-stimulation of the mucosal immune response, especially in diets formulated with very low FM levels (2%), allowing more of the available nutrients and energy to be used for growth and other productive functions rather than being diverted to the immune response.

Recently, different plasma metabolites considered as biomarkers of the fish physiological status such as glucose, lactate, and total protein, as well as cortisol levels, have been correlated with the skin mucus levels in several fish species subjected to a stress challenges [39,46,47,48,49,50]. These classic biomarkers measured in skin mucus could also report the fish response to fasting or pathogenic infections [20]. Interestingly, our data indicate that the SDPP diet with low FM levels did not alter the general skin mucus composition, as it was recently demonstrated in the SDPP diet with high FM levels [18] or in other functional diets [14,19,51]. Although studies that relate the nutritional status with the levels of metabolites in the skin mucus are still scarce, everything seems to indicate that the SMABs composition could be essential for the proper functioning of the skin mucus barrier and a reliable indicator of fish condition [20,39,49]. In this study, SMABs levels were within the normal range of values reported in other studies under different dietary conditions [14,20,39,51], which indicates that no effect exists due to the inclusion of SDPP in the diet. Beyond the main SMABs, cortisol levels are of major interest in fish studies. This steroid is the main hormone, secreted and released, and involved in the stress response. The cortisol response can strongly vary within species depending on the severity or duration of the stressor [52]. Although the mechanisms involved in cortisol exudation are still unknown, skin mucus cortisol has been detected in fish [53,54] and its correlation levels between mucus and plasma has been recently confirmed in meagre [48]. Moreover, the above-mentioned works were conducted under acute stress, where the authors showed that cortisol release levels were dependent on the stressor’s nature. In addition, the reduction in cortisol levels would also favor the fish’s mucosal immunity, promoting barrier robustness and therefore pathogen resistance [41]. Similar results were obtained in gilthead seabream fed a phytogenic supplemented diet [51]. Under the current conditions with very low FM diets, the drastic reduction in exuded cortisol from fish fed the SDPP diet in comparison to the control group could indicate a decrease in stress levels, which highlights the physiological benefits of the tested functional diet.

In the present study, the skin mucus antibacterial capacity in response to dietary conditions was also evaluated. The pathogenic strains selected in this work, *V. anguillarum* and *P. anguilliseptica* are well-characterized as causing vibriosis and pseudomoniasis in several species [55]. *E. coli* has been used as indicative control of the potential skin mucus antibacterial capacity without previous pathogenic bacterial strains contact of used specimens that could generate specific acquired defenses. In addition, skin mucus incorporates a reservoir of numerous innate immune components and antimicrobial peptides, as well as immunoglobulins, which exert inhibitory or lytic activity against pathogenic microorganisms [56]. Taking into consideration that mucus properties are crucial in the adherence of bacteria to mucosal surfaces and the subsequent infection [57], and as the mucus is continuously generated and exuded with a rapid renewal process [58,59], developing diets that may improve the antibacterial potential of the fish mucus can provide relevant benefits during fish farming. Confirming the immunomodulatory properties of the tested ingredient was demonstrated by the ex vivo trial, showing an evident higher inhibitory capacity of the skin mucus of fish fed the SDPP against *V. anguillarum* in the co-culture challenges, whereas the benefits against *P. anguilliseptica* were raised on both diets. Furthermore, authors have also evidenced that dietary SDPP promoted skin development by increasing the thickness of the epidermis and the *stratum spongiosum* of the dermis, providing new evidence of the benefits of this ingredient on the skin [18]. In another mucosal tissue, several authors have reported that the inclusion of SDPP in animal diets (mice, pigs, and sea bream) modified the intestinal microbiota, for example increasing the *Lactobacillaceae* family [60,61,62], a well-known and recognized commensal bacterial group associated with health benefits in animals and by extension in humans. These results have also been found in fish fed SDP [10,12,16], and may be attributed to the protein-like components (peptides and free amino acids profiles) of this ingredient [63].

The above-mentioned results in response to the SDPP diet, confirm a dietary systemic effect through the immunomodulatory capacity of the tested diets on primary splenocyte cell cultures exposed to bacterial LPS, and a diet local effect through the antibacterial capacity of skin mucus co-cultured. In addition, the beneficial effects on skin mucus in response to protein hydrolysates had also been confirmed in meagre [19] and seabream [14,18], benefits that could be imputed to the bioactive compounds existing in SDPP, such as growth factors, albumins, immunoglobulins, and biological active peptides, which may moderate the immunomodulatory and anti-inflammatory effects [64,65]. These dietary systemic and mucosal effects confirmed the prior evidence of the metabolic, immune, and protein transport processes that occur in response to SDPP at the mucosal tissue level [15,18]. In similar studies conducted by Vallejos et al. [15] and Reyes-López et al. [18], SDPP was tested at higher levels of FM inclusion (47%), and the results from the above-mentioned studies indicated a remarkable association between immunity, protein transport, and metabolic-associated processes in response to the SDPP diet. This evidence confirmed the previous manifest of the mucosal health enhancement status, and metabolic and immunology processes in response to the SDPP dietary inclusion and provides further understanding of the method of action of this ingredient in aquafeeds.

## 5. Conclusions

Genes involved in humoral immunity (*IgM*), along with pro- (*il-1β*), and anti-inflammatory (*tgf-β1*) cytokines, and the surface cell marker cd4 (*cd4*) were up-regulated in primary cell cultures of splenocytes exposed to LPS in fish fed the SDPP diet in comparison to the control diet, which indicated that this functional ingredient was able to modulate systemic immune response when exposed to a bacterial-type PAMP. Furthermore, main SMABs were not modified by the SDPP diet, and even skin mucus cortisol levels were drastically reduced, which could be considered beneficial from a physiological perspective. Skin mucus plays an important role in host defense, particularly in the prevention of colonization by pathogenic microorganisms. This defense can be modified by the SDPP diet broadly reducing the settlement of the studied bacteria on the skin surface, decreasing the risk of infection.

Results from this study indicated that animal by-products, especially in the current very low FM diets, play an important role in the aquaculture industry. Spray-dried porcine plasma seems to be useful and beneficial at the immunological level and mucosal health status, which makes it a good candidate for incorporation into aquaculture.

## Figures and Tables

**Figure 1 animals-12-03297-f001:**
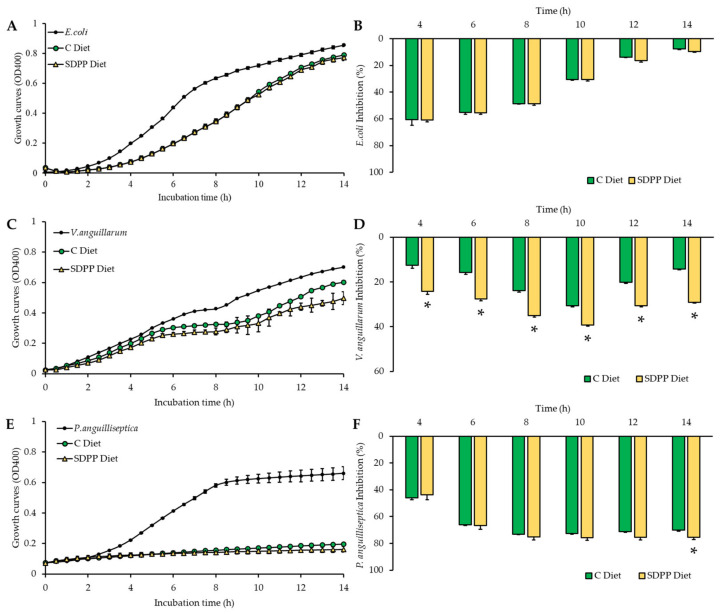
Skin mucus bacterial growth and inhibition rate. The bacterial growth (**left**) and the bacterial inhibition rate (**right**) are shown for (**A**,**B**) *E. coli*, (**C**,**D**) *V. anguillarum*, and (**E**,**F**) *P. anguilliseptica* co-cultured for 14 h with gilthead seabream (*Sparus aurata*) skin mucus from fish fed with the control diet (C Diet) and SDPP Diet. Data are shown as mean ± standard error of the mean (*n* = 9). An asterisk (*) indicates statistically significant differences between diets at measured times of co-culture (*t*-test; *p* < 0.05).

**Figure 2 animals-12-03297-f002:**
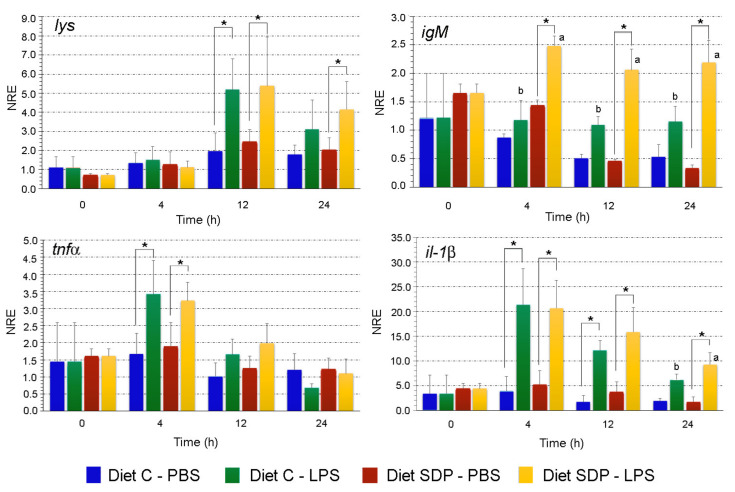
Gene expression analysis of humoral innate response markers measured in primary cultures of splenocytes incubated with PBS or LPS. The gene expression (*n* = 6) of lysozyme (*lys*), immunoglobulin M (*igM*), tumor necrosis factor alpha (*tnfα*), and interleukin 1β (*il-1β*) from splenocytes of gilthead sea bream (*Sparus aurata*) fed with a control diet (Diet C) with very low levels of fishmeal (7%) and a diet in which 5% fishmeal was replaced with spray-dried porcine plasma (SDPP diet). Gene expression was measured in primary cultures of splenocytes from both groups exposed to PBS and LPS at different sampling times (0, 4, 12, and 24 h after LPS or PBS incubation). The analysis used the β-actin as the reference gene. Data were compared by means of a two-way ANOVA. Different letters indicate statistically significant differences between splenocytes from different experimental groups (C and SDPP diets) at the same sampling time. The asterisk indicates the presence of statistically significant differences between splenocytes incubated with PBS or LPS within the same experimental group (*p* < 0.05).

**Figure 3 animals-12-03297-f003:**
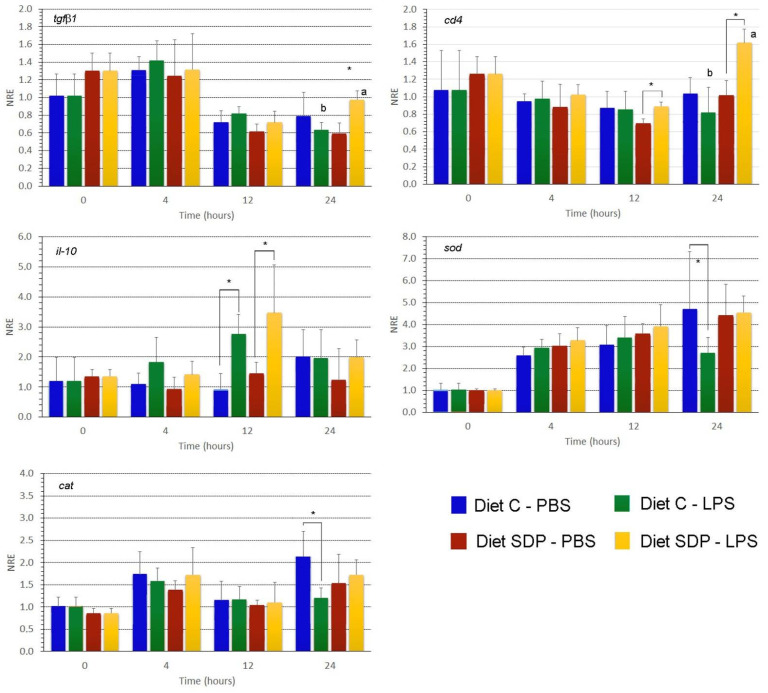
Gene expression analysis of anti-inflammatory, surface, and anti-oxidative markers measured in primary cultures of splenocytes incubated with PBS or LPS. The gene expression (*n* = 6) of anti-inflammatory cytokines (*il-10*, *tgfβ1*), the surface marker *cd4* and antioxidative stress enzymes (*sod*, *cat*) was evaluated in splenocytes of gilthead sea bream (*Sparus aurata*) fed a control diet (Diet C) with very low levels of fishmeal (7%) and a diet in which 5% fishmeal was replaced with spray-dried porcine plasma (SDPP diet). The gene expression profile was measured in primary cultures of splenocytes from both groups exposed to PBS and LPS at different sampling times (0, 4, 12 and 24 after LPS or PBS incubation). The analysis used the β-actin gene as the reference. Data were compared by means of a two-way ANOVA. Different letters denote statistically significant differences between splenocytes from different groups (C and SDPP diets) at the same sampling time. The asterisk indicates the presence of statistically significant differences between splenocytes incubated with PBS or LPS within the same experimental group (*p* < 0.05).

**Table 1 animals-12-03297-t001:** List of ingredients and proximal composition of experimental diets used for evaluating the nutritional and functional properties of spray-dried porcine plasma (SDPP) on gilthead sea bream (*Sparus aurata*) juveniles.

Ingredients (%)	Control Diet	SDPP Diet
Fish meal LT70 (Norvik)	7.0	2.0
Appetein GS (APC Europe)	-	5.0
Soy protein concentrate (Soycomil P)	21.0	21.0
Pea protein concentrate (Lysamine GPS)	12.0	12.0
Wheat gluten (VITAL)	12.0	12.0
Corn gluten	12.0	12.0
Soybean meal 48	5.0	5.0
Wheat meal	10.4	10.4
Fish oil (SAVINOR)	15.0	15.0
Vitamin and mineral (Premix PV01)	1.0	1.0
Soy lecithin–Powder	1.0	1.0
Binder (guar gum)	1.0	1.0
Monocalcium phosphate (ALIPHOS MONOCAL)	2.0	2.0
L-Lysine	0.3	0.3
L-Tryptophan	0.1	0.1
DL-Methionine (Rhodimet NP99)	0.2	0.2
Total	100.0	100.0
Crude protein, %	48.4	48.7
Crude fat, %	17.2	17.0
Fiber, %	1.5	1.5
Ash, %	5.9	5.5
Gross Energy, MJ/kg	21.6	21.7

**Table 2 animals-12-03297-t002:** Growth and feed key performance indicators of gilthead sea bream (*Sparus aurata*) juveniles fed low fishmeal diets containing 5% of spray-dried porcine plasma (SDPP).

	Control Diet	SDPP Diet
Survival (%)	96.8 ± 2.7	98.0 ± 1.5
Standard length (SL, cm)	18.7 ± 0.2	19.0 ± 0.3
Body weight (BW, g)	173.8 ± 6.0	180.0 ± 4.4 *
Specific growth rate (SGR, %BW/day)	2.06 ± 0.03	2.10 ± 0.01 *
Fulton’s condition factor (K)	2.65 ± 0.1	2.64 ± 0.2
Feed conversion ratio (FCR)	0.87 ± 0.06	0.90 ± 0.11

Data are shown as the mean ± standard deviation (SD). The asterisk (*) denotes statistically significant differences between both groups (*t*-test, *p* < 0.05; *n* = 40).

**Table 3 animals-12-03297-t003:** Skin mucus biomarkers and their ratios from gilthead sea bream (*Sparus aurata*) juveniles fed low fishmeal diets containing 5% of spray-dried porcine plasma (SDPP).

	Control Diet	SDPP Diet
**Skin mucus biomarkers**		
Glucose (µg/mL)	14.98 ± 2.06	16.13 ± 1.36
Lactate (µg/mL)	7.24 ± 1.06	7.03 ± 0.68
Protein (mg/mL)	9.01 ± 1.20	10.00 ± 1.72
Cortisol (ng/mL)	0.40 ± 0.20	0.23 ± 0.06 *
FRAP (µmol/mL)	1387 ± 105	1347 ± 63
**Skin mucus ratios**		
Glucose/Protein (µg/mg)	1.85 ± 0.24	1.74 ± 0.21
Lactate/Protein (µg/mg)	0.82 ± 0.06	0.76 ± 0.10
Cortisol/Protein (ng/g)	53.4 ± 25.8	32.5 ± 7.9 *
FRAP/Protein (µmol/mg)	154 ± 10	124 ± 10 *
Glucose/Lactate (mg/mg)	2.41 ± 0.15	2.12 ± 0.17

Values are mean ± standard deviation (SD) of the mean from ten individual fish. The asterisk (*) denotes statistically significant differences between both diets (*t*-test; *p* < 0.05; *n* = 40).

## Data Availability

Not applicable.

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
