# Peer review of "Evaluating the Functional Properties of Spray-Dried Porcine Plasma in Gilthead Seabream (Sparus aurata) Fed Low Fish Meal Diets"

_animals, 2022, doi:10.3390/ani12233297_

Round 1

Reviewer 1 Report

The MS reports novel results about the benefits of SDPP in livestock diets and especially on the immunomodulatory effects of its dietary inclusion in aquatic feeds. I have included my comments (mostly related to the use of some statistic tests and to the low quality of the figures that should be improved) in the pdf attached 

Author Response

The MS reports novel results about the benefits of SDPP in livestock diets and especially on the immunomodulatory effects of its dietary inclusion in aquatic feeds. I have included my comments (mostly related to the use of some statistic tests and to the low quality of the figures that should be improved) in the pdf attached 

(x) I don't feel qualified to judge about the English language and style

Authors would like to inform this reviewer and the Editorial board that among one of the coauthors, there is K.B. Andree who is a native English speaker. Regardless of it, the text has been revised again and typographic errors have been corrected when found.

Results:

Line 268-269: add p valor

R: The P value has been added

Fig 2: please provide a clearer fig 2. Be sure to provide at least the same scale scales for all graphics (example the third is too different in scale from the others).

R: New figures have been provided. Figures have a resolution of 300 dpi so this fits the image quality requirements for publication.

Fig 3: Same comment that fig 2. It is not clear and all the imagines use a different scale. Please revise. Have you tested the normality of data before to use the ANOVA? (line 372).

R: Authors acknowledge the comment from the reviewer, but we would like to state that in the original version of our manuscript, this information was already stated (“All data were checked for normality and homoscedasticity prior their analysis”). Regarding the scales in figures 2 and 3, we prefer to keep the scales as they are in the original submission, since this allows to clearly visualize the results for each gene, whereas if we change the scale, we will lose this feature. In addition, uniformizing graphs scales is not relevant since we do not compare gene expression profiles among genes.

Reviewer 2 Report

The authors evaluated the functional properties of spray-dried porcine plasma in gilthead seabream (Sparus aurata) fed low fish meal diets. They designed two treatments to test their hypothesis. This manuscript (MS) was clearly written and easy to understand. They covered a wide range of factors, from growth to gene expression. This work can help the sustainability of this species farming as few studies have been done on this topic. However, some major issues significantly compromised the quality of this MS.

Major comments:

  • First, the manuscript needs to be edited by a native English speaker to improve the language of the MS and fix errors.
  • Very poor literature search and many parts of the discussion should be revised.

However, I have touched on some more points that can contribute to the improvement of this MS.

Minor comments

Abstract

·       Line 21-23, please revise it. Further, please mention due to having what?

·       Line 23, local??

·       Line 25, please delete “biomarkers” throughout the MS. The concept of biomarkers is a bit broad and tricky.

·       Line 35, the ingredient is not correct; please change it to compound.

·       Line 42, please clarify what you mean by “ex vivo”.

·       Here and elsewhere, report P uppercase and italic (P<0.05).

·       Throughout the MS, if there is no significant difference, no need to report P-value.

·       Please reorder the keywords alphabetically and capitalize each word.

·       Please write the abstract more numerically about the results. You can do it by adding their numbers in parentheses.

Introduction:

•     Well-developed introduction and included a clear fellow and relevant points.

·       Here and throughout the MS, please first mention the common name plus scientific name, and for the rest of the MS, just report the common name.

·       Please update the introduction with recent works as many studies are available from the last two years, which were not included in this section.

·       Please review the literature much more carefully and cite more appropriate references. Please check any single reference by reading it instead of citing it without knowing about its concepts.

·       Line 55, the functional feed that you mentioned, is a bit different than this study. Please delete it.

·       Line 70-71, is not correct; This ingredient is good due to its rich source of amino acids and not an antibiotic. Please search the literature better.

·       Line 75, does not improve feed intake and feed efficiency. Please read more of the literature; even the problem with this ingredient is low palatability.

·       Please delete the “local” throughout the MS.

·       Please mention the novelty of your work in the last paragraph of the introduction.

Material and methods

·       Well-organized section. Clear fellow and all required details were provided.

·       Table 1, please provide information regarding the used ingredients in the footnote of this table.

·       Line 257, why? Please be consistent with SD or SE.

·       Line 259, The analysed two-way is not correct. Please see this article and others to see how you should report two-way analysis (https://www.hindawi.com/journals/anu/2022/8991678/). Please cite the reference for validation.

·       For each analysis, please clarify how many fish were taken.

Results

·       Well-written section, all necessary things have been covered but need to be more numeric.

·       Line 267, 3% is nothing; please do not focus too much on improved growth.

·       Table 2, was the FCR below 1!!!. Please double-check this data.

·       Line 271 and elsewhere, please delete the biomarkers term throughout the MS.

Discussion

         Line 390-391, please delete this part. Please read my comment regarding the functional feeds.

         Line 391-396, you mentioned earlier in the introduction. Please focus on the pros and cons of this ingredient and study more.

         Line 410-403, please focus on blood meal studies as hips of references are available.

         Line 443-446; are completely different studies. Please only focus on the parameters you measured in this study.

         Line 456-469. Please consider that they measure cortisol in blood and plasma and not mucus. Please find the references that they measured on mucus and then discussed here.

         You did not measure the contents of many compounds that you discussed here. You have to measure them or only focus on the parameters that you measured in this study.

         As a general comment: please focus on fish as hips of the references and studies are available, and no need to cite other vertebrates.

·       Some parts of the discussion are better updated with research in 2021 and 2022 as they refer to some old references. Please update the discussion with the latest studies as much as possible.

·       Although you wrote this section well, you can still improve it by answering these questions and annotating them in the discussion section. Why were these results observed? Discuss more possible reasons.

·       The conclusion needs to be revised and more comprehensive concepts should be added there.

Tables and Figures

•            Please explain a little bit about your experimental diets, per each Table and Figure. Each Table and figure should represent enough information separately from the text.

•            Double-check the units and titles of all Tables.

•            Please mention in the footnote of all Tables which kind of statistical method you used for comparing the means.

When revising your manuscript, please consider all issues mentioned in the reviewers' comments carefully with clear outlines for every change made in response to their comments including suitable rebuttals for any comments you deem inappropriate. Please itemize your response to each review comment and highlight the revised at re-submission.

Kind regards

Author Response

Major comments:

  • First, the manuscript needs to be edited by a native English speaker to improve the language of the MS and fix errors.

R: Authors would like to inform this reviewer and the Editorial board that among one of the coauthors, there is K.B. Andree who is a native English speaker. Regardless of it, the text has been revised again and typographic errors have been corrected when found.

  • Very poor literature search and many parts of the discussion should be revised.

R: We appreciate reviewer’s comment, but literature has been deeply scrutinized and all published articles dealing with spray dried plasma (SDP) are cited and deeply discussed in our contribution. We would like to clarify that our manuscript does not deal with hemoderivates as alternative sources to fishmeal, this study is focused on SDP as a functional ingredient, and for this reason no citations devoted to blood meals as alternative protein sources have not been considered in our manuscript.

However, I have touched on some more points that can contribute to the improvement of this MS.

Minor comments

Abstract

  • Line 21-23, please revise it. Further, please mention due to having what?

R: This sentence has been rewritten following the advice of the reviewer in order to highlight the benefits of spray-dried porcine plasma as a functional ingredient; since it does not only promote somatic growth but it also has beneficial effects in terms on stress and immune responses in fish, as it is presented and discussed in the manuscript.

  • Line 23, local??

R: Following the comment of the reviewer, the term “local” has been replaced by “mucosal”.

  • Line 25, please delete “biomarkers” throughout the MS. The concept of biomarkers is a bit broad and tricky.

R: Authors acknowledge the comment from the reviewer, but a biomarker is generally a chemical substance used as an indicator of biological status. As it was discussed in previous studies, the variation in skin mucus protein could indicate an infection; a variation in glucose levels could indicate a stress or a fasting situation; a lactate variation levels could indicate a change in metabolism pathways; and FRAP is an indicator of oxidative stress. Due to the above mentioned and also described in several papers, the term biomarker in this manuscript has been properly used.

  • Line 35, the ingredient is not correct; please change it to compound.

R: According to regulations of feed manufacture, SDP is considered as an ingredient in feed formulations.

  • Line 42, please clarify what you mean by “ex vivo”.

R: An ex vivo refers to experimentation or measurements performed on tissues in an artificial environment outside the organism with minimal alteration of natural conditions. In our case, we use spleens from animals feed both diets and run cultivated them on cell culture media to evaluate their response to LPS. As the definition of an ex vivo assay in comparison to an in vitro trial is well known in the literature, authors do not consider necessary to describe it in our paper.

  • Here and elsewhere, report P uppercase and italic (P<0.05).

R: Thanks for the reviewer’s valuable comment. Changes have been made throughout the manuscript.

  • Throughout the MS, if there is no significant difference, no need to report P-value.

R: Regardless of reviewer comment, we keep (P > 0.05) following journal’s guidelines.

  • Please reorder the keywords alphabetically and capitalize each word.

R: Thanks for the reviewer’s suggestion. The change has been done.

  • Please write the abstract more numerically about the results. You can do it by adding their numbers in parentheses.

R: We appreciate this comment but considering the limited extension of the Abstract in terms of number of words, there is not enough space for properly describing our results. Furthermore, including numerical data does not change the final message which is the promoting of growth and modulation of the immune response in fish. Adding numerical values will not change substantially this information but will compromise the clarity of the final message included in the Abstract. For this reason, authors decided not to implement reviewer’s comment in this revised version of the MS.

Introduction:

Well-developed introduction and included a clear fellow and relevant points.

  • Here and throughout the MS, please first mention the common name plus scientific name, and for the rest of the MS, just report the common name.

R: This has been checked and corrected when needed.

  • Please update the introduction with recent works as many studies are available from the last two years, which were not included in this section.

R: As we stated, all published studies dealing with spray dried plasma as a functional ingredient have been included in this manuscript, and we can be quite certain on this, since most of this works were published by our research groups. In this sense, the MS does not deal with blood meal as an alternative protein source, rather than as a functional ingredient, for this reason no literature is cited on studies dealing with different hemoderivates as alternative proteinic sources as this is not the topic addressed by the manuscript. In general: 80% are citations from the last 10 years, and 50% are citations from the last 5 years.

  • Please review the literature much more carefully and cite more appropriate references. Please check any single reference by reading it instead of citing it without knowing about its concepts.

R: This issue has been already addressed in our previous answer.

  • Line 55, the functional feed that you mentioned, is a bit different than this study. Please delete it.

R: Different functional feeds may have similar beneficial effects on the host, so it is relevant to support our findings with another study even though that study was conducted with another additives, since what we are seeking is to support its functional benefits citing similar results found in other nutritional studies regardless of using different dietary strategies.

  • Line 70-71, is not correct; This ingredient is good due to its rich source of amino acids and not an antibiotic. Please search the literature better.

R: We have rewritten this sentence in order to avoid misunderstandings from the reader side.

  • Line 75, does not improve feed intake and feed efficiency. Please read more of the literature; even the problem with this ingredient is low palatability.

R: We do not agree with the reviewer, SDP is reputed for its good palatability as it has been reported in livestock and aquatic species, so we do not understand the comment of the reviewer. For instance, SDP enhances feed intake in swine during weaning due to its good palatable properties. In fish these effects are not so evident, but they are also positive and in other studies from our group, we have found an improvement in FCR values (Gisbert et al., 2015), although not in this study, which may be due to the low values of fishmeal inclusion in the diet.

  • Please delete the “local” throughout the MS.

R: The term “local” has been replaced by “mucosal”.

  • Please mention the novelty of your work in the last paragraph of the introduction.

R: The novelty of this work is the proof of this ingredient in diets with only 2% fish-meal. As it was described in the last paragraph of the introduction:

Thus, using as a biological model the gilthead seabream, the objective of this study was to evaluate the beneficial effects of SDPP in diets formulated with very low fish-meal (2%) in terms of standard key performance indicators like somatic growth and feed efficiency parameters and test whether diets containing SDPP promoted the systemic immune response in gilthead sea bream by: a) a mucosal immune response through challenging skin mucus antibacterial capacity in order to evaluate the response of the skin-associated lymphoid tissue in front of bacterial pathogens; b) a systemic immune response through their splenocytes ex vivo with a pathogen-associated molecular pat-tern like lipopolysaccharide (LPS) and evaluate the effects of the functional ingredient on the systemic immune response of the host.

 Material and methods

Well-organized section. Clear fellow and all required details were provided.

  • Table 1, please provide information regarding the used ingredients in the footnote of this table.

R: This information has been added to the new revised version following the advice of the reviewer.

  • Line 257, why? Please be consistent with SD or SE.

R: All results were expressed as mean ± standard deviation (SD). Only the skin mucus antibacterial activity was expressed as mean ± standard error (SE) due to this is done in microbiology for a better visualization of the data.

  • Line 259, The analysed two-way is not correct. Please see this article and others to see how you should report two-way analysis (https://www.hindawi.com/journals/anu/2022/8991678/). Please cite the reference for validation.

R: More information about the Two-way ANOVA has been included in the results section, as well as including the P value for the interaction between both factors

  • For each analysis, please clarify how many fish were taken.

R: This information is included in M&M section and in the figure captions.

Results

Well-written section, all necessary things have been covered but need to be more numeric.

  • Line 267, 3% is nothing; please do not focus too much on improved growth.

R: We agree that 3% is not very important in a small trial like ours, it is a relevant result that we need to discuss. Following reviewer’s advice, we have put more emphasis on the other sections of the MS (mucus analyses and ex-vivo gene expression results).

  • Table 2, was the FCR below 1!!!. Please double-check this data.

R: Getting lower FCR values than 1 is not so uncommon as reviewer states, due to improvements in feed formulation and high-quality ingredients coupled to optimal rearing conditions, lower FCR than 1 may be found. This is not so rare, other studies dealing with salmonids have found the same trend when running trials under optimal conditions.

  • Line 271 and elsewhere, please delete the biomarkers term throughout the MS.

R: This issue has already been discussed previously. Nevertheless, a biomarker is generally a chemical substance used as an indicator of biological status, so in this case, as it was published previously in other studies, glucose, lactate, protein, cortisol and also FRAP are indicators of biological status.

Discussion

  • Line 390-391, please delete this part. Please read my comment regarding the functional feeds.

R: Above discussed.

  • Line 391-396, you mentioned earlier in the introduction. Please focus on the pros and cons of this ingredient and study more.

R: Above discussed.

  • Line 410-403, please focus on blood meal studies as hips of references are available.

R:  The MS does not deal with blood meal as an alternative protein source, rather than as a functional ingredient, for this reason no literature is cited on studies dealing with different hemoderivates as alternative proteinic sources as this is not the topic addressed by the manuscript.

  • Line 443-446; are completely different studies. Please only focus on the parameters you measured in this study.

R: When discussing our results, it is important to provide a clear interpretation and discussion of the results by citing relevant studies, if different studies did find similar results than ours, they may serve for supporting our results and provide strong evidence of the positive effects of the ingredient. For this reason, authors decide to keep their style when writing the discussion as it is.

  • Line 456-469. Please consider that they measure cortisol in blood and plasma and not mucus. Please find the references that they measured on mucus and then discussed here.

R: Sorry if we have not understood the question but we will try to answer along with the text:

¨Cortisol is secreted and released in responses to stress. The cortisol levels can strongly vary within species according to the duration or severity of the stressor [54] (the variability is in general: plasma, mucus or any tissue). Although the mechanisms involved in cortisol exudation are still unknown, skin mucus cortisol has been detected in fish [55,56] (mucus studies) and the correlation between plasma cortisol and mucus cortisol has been recently confirmed in meagre [50] (skin mucus study). Moreover, the above-mentioned works were performed under short-term acute stress and the authors stated that cortisol response could be dependent on the stressor. In addition, the reduction in cortisol levels would also favor the fish’s local mucosal immunity, promoting more effective defense responses against pathogens [43] (mucosal study). Similar results were obtained in gilthead seabream fed a phytogenic supplemented diet [53] (plasma and mucus study). Under the current conditions, the drastic reduction of exuded cortisol from fish fed the SDPP diet in comparison to the control group could indicate a decrease in stress levels, which highlights the physiological benefits of the tested functional diet¨.

  • You did not measure the contents of many compounds that you discussed here. You have to measure them or only focus on the parameters that you measured in this study.

R: Sorry, we would appreciate knowing which compounds you are referring to. If any have been discussed to broaden the discussion, it has always been based on published references to understand the results obtained. Also, we don't know what this comment refers to.

  • As a general comment: please focus on fish as hips of the references and studies are available, and no need to cite other vertebrates.

R: We disagree with the reviewer once again, results from livestock studies are really interesting for supporting and providing further and strong evidence that the results found by authors are not isolated and they are common for other species. Comparing results with other vertebrates gives a more solid conclusion of the results and their potential impact. This is a common strategy in many studies, especially those working with ingredients already tested in livestock where there is a bulk on relevant literature on them.

  • Some parts of the discussion are better updated with research in 2021 and 2022 as they refer to some old references. Please update the discussion with the latest studies as much as possible.

R: All cited works are cited according to their importance and relevance in the field rather than their date of publication. The relevance of a study does not depend on its date of publication, rather than the importance and relevance of the published results.

  • Although you wrote this section well, you can still improve it by answering these questions and annotating them in the discussion section. Why were these results observed? Discuss more possible reasons.

R: Thanks for the reviewer’s valuable comments. It has been modified.

  • The conclusion needs to be revised and more comprehensive concepts should be added there.

R: Thanks for the reviewer’s valuable comments. It has been modified.

Tables and Figures

  • Please explain a little bit about your experimental diets, per each Table and Figure. Each Table and figure should represent enough information separately from the text.

R: All tables and figure captions were:

Table 1. List of ingredients and proximal composition of experimental diets used for evaluating the nutritional and functional properties of spray-dried porcine plasma (SDPP) on gilthead sea bream (Sparus aurata) juveniles.

Table 2. Growth and feed key performance indicators of gilthead sea bream (Sparus aurata) juveniles fed low fishmeal diets containing 5% of spray-dried porcine plasma (SDPP). Data are shown as the mean ± standard deviation (SD). The asterisk (*) denotes statistically significant differences between both groups (t-test, P < 0.05; n=40).

Table 3. Skin mucus biomarkers and their ratios from gilthead sea bream (Sparus aurata) juveniles fed low fishmeal diets containing 5% of spray-dried porcine plasma (SDPP). Values are mean ± standard deviation (SD) of the mean from ten individual fish. The asterisk (*) denotes statistically significant differences between both diets (t-test; P < 0.05; n=40).

Figure 1. Bacterial growth and inhibition rate in the skin mucus from fish fed with the SDPP diet. The bacterial growth (left panels) and the bacterial inhibition rate (right panels) are showed for (A, B) E. coli, (C, D) V. anguillarum, and (E, F) P. anguilliseptica co-cultured for 14 h with gilthead seabream (Sparus aurata) skin mucus from fish fed with both experimental diets (control and SDPP diets). Data are shown as the mean ± standard error of the mean (n = 9). An asterisk (*) denotes statistically significant differences between both diets at measured times of co-culture (t-test; P < 0.05).

Figure 2. Gene expression analysis of humoral innate response markers measured in primary cultures of splenocytes incubated with PBS or LPS. The gene expression (n = 6) of lysozyme (lys), immunoglobulin M (igM), tumor necrosis factor alpha (tnfα), and interleukin 1β (il-1β) from splenocytes of gilthead sea bream (Sparus aurata) fed with a control diet (Diet C) with very low levels of fishmeal (7%) and a diet in which 5% fishmeal was replaced with spray-dried porcine plasma (SDPP diet). Gene expression was measured in primary cultures of splenocytes from both groups exposed to PBS and LPS at different sampling times (0, 4, 12, and 24 h after LPS or PBS incubation). The analysis used the β-actin as the reference gene. Data were compared by means of a two-way ANOVA. Different letters denote statistically significant differences between spleno-cytes from different experimental groups (C and SDPP diets) at the same sampling time. The asterisk indicates the presence of statistically significant differences between splenocytes incubated with PBS or LPS within the same experimental group (P < 0.05).

Figure 3. Gene expression analysis of anti-inflammatory, surface, and anti-oxidative markers measured in primary cultures of splenocytes incubated with PBS or LPS. The gene expression (n = 6) of anti-inflammatory cytokines (il-10, tgfβ1), the surface marker cd4 and antioxidative stress enzymes (sod, cat) was evaluated in splenocytes of gilthead sea bream (Sparus aurata) fed a control diet (Diet C) with very low levels of fishmeal (7%) and a diet in which 5% fishmeal was replaced with spray-dried porcine plasma (SDPP diet). The gene expression profile was measured in primary cultures of splenocytes from both groups exposed to PBS and LPS at different sampling times (0, 4, 12 and 24 after LPS or PBS incubation). The analysis used the β-actin as the reference gene. Data were compared by means of a two-way ANOVA. Different letters denote statistically significant differences between splenocytes from different groups (C and SDPP diets) at the same sampling time. The asterisk indicates the presence of statistically significant differences between splenocytes incubated with PBS or LPS within the same experimental group (P < 0.05).

  • Double-check the units and titles of all Tables.

R: Thanks for the reviewer’s valuable comments. It has been modified.

  • Please mention in the footnote of all Tables which kind of statistical method you used for comparing the means.

R: Sorry if we have not understood but in table 2 and 3 the statistical methods are indicated (t-test). The table 1 it’s the List of ingredients.

Author Response

  1. x) English language and style are fine/minor spell check required

R: Authors would like to inform this reviewer and the Editorial board that among one of the coauthors, there is K.B. Andree who is a native English speaker. Regardless of it, the text has been revised again and typographic errors have been corrected when found.

Title is appropriate as per the content

abstract: line no 47-48 seems to be contradictory as cholesterol, synthesis pathways and also a part of the weight gain in fish. Can you justify it?

R: Sorry but in this manuscript we do not investigate cholesterol or their synthesis pathways.

Introduction: authors are advised to put the latest references if possible 80 % should be 2016 onwards.

R: The literature has been reviewed for all authors: 80% are citations from the last 10 years, and 50% are citations from the last 5 years.

The author should refer another researcher who has invested other non-conventional feed ingredients. The following references should be put where ever applicable:

  • Manish Jayant, Narottam Prasad Sahu, Ashutosh Dharmendra Deo, Subodh Gupta, Kooloth Valappil Rajendran, Chetan Kumar Garg, Dharmendra Kumar Meena, Minal Sheshrao Wagde,Effective valorization of bio-processed castor kernel meal based fish feed supplements concomitant with oil extraction processing industry: A prolific way towards greening of landscaping/environment, Environmental Technology & Innovation, 21, 2021,101320.
  • Jayant, M.A. Hassan, P.P. Srivastava, D.K. Meena, P. Kumar, A. Kumar, M.S. Wagde, Brewer’s spent grains (BSGs) as feedstuff for striped catfish, Pangasianodon hypophthalmus fingerlings: An approach to transform waste into wealth, Journal of Cleaner Production, Volume 199, 2018, Pages 716-722.

R: There might be an error from the file submitted by the reviewer, since these two references are out of context considering the topic of our manuscript.

Material and methods

Authors has used only two treatments including control. My concern is whether author has assessed the range finding. How author decided that a single inclusion is sufficient for deriving their conclusion.

R: Due to the bibliography and the knowledge of APC Europe SL, in 2015 a study showed the best SDPP inclusion that promotes growth in Sea bream (Gisbert et al, 2015).

Results

Table 2 can be extended with more parameters like WG %, ADG, WG, PER, FER, HSI, GaSI, CSI, etc.,while one has basic information.

R: It is not relevant for authors include information on the recommended variables, our work is basically on the effects of SDP on fish condition measured by means of mucus biomarkers and immune response of fish by means of an ex vivo trial. Data on growth and FCR are just for illustrating the positive effects of the ingredient of fish that support the good results found in the rest of physiological variables analysed.

References

Author must follow the format of the journal.

R: The reference format has been made with the Mendeley program following the Animals format.

Round 2

Reviewer 2 Report

Please look at the comment one more time. You did not address some of them. For example, literature search is required many works. Please get back to the comments one by one and address them.

Author Response

We thank the reviewer for his/her comment, however, we disagree with him/her.

As far as we are concerned we addressed all queries indicated by this reviewer, especially those related to the statistics and cited literature. We would like to highlight once again that the manuscript does not deal on blood meal as a fishmeal substitute, our manuscript is focused on the functional properties of spray-dried plasma in terms of immunity. For this reason, we do not consider relevant to include literature on blood meals as alternative sources of protein, as it is out of context.

Would the reviewer be more precise on the issues that we have not addressed?

Thanks